


# Impact of a long-lived anticyclonic mesoscale eddy on seawater anomalies in the northeastern tropical Pacific Ocean: A composite analysis from hydrographic measurements, sea level altimetry data and reanalysis model products

Kaveh Purkiani[1,2], Matthias Haeckel[2], Sabine Haalboom[3], Katja Schmidt[4], Peter Urban[2,5], Iason-Zois Gazis[2], Henko de Stigter[3], André Paul[1], Maren Walter[1], and Annemiek Vink[4]

[1]MARUM Center for Marine Environmental Sciences and Faculty of Geosciences, University of Bremen, Bremen, Germany.
[2]GEOMAR Helmholtz Center for Ocean Research Kiel, Kiel, Germany
[3]NIOZ Royal Netherlands Institute for Sea Research, Department of Ocean Systems, Texel, The Netherlands
[4]Federal Institute for Geosciences and Natural Resources (BGR), Hannover, Germany
[5]Department of Geology, Ghent University, Ghent, Belgium

**Correspondence:** Kaveh Purkiani (kpurkiani@marum.de)

**Abstract.** Using observational data, satellite altimeters, and reanalysis model products, we have investigated eddy-induced seawater anomalies and heat/salt transport in the northeastern tropical Pacific Ocean. An eddy detection algorithm (EDA) was used to identify eddy formation at the Mexican Tehuantepec Gulf (TT) in July 2018 during an unusually strong summer wind event. The eddy separated from the coast with a mean translation velocity of 11 $\mathrm{cm\,s^{-1}}$ and a mean radius of 115 km and travelled 2050-2400 km westwards off the Central American coast, where it was followed at approx -114°W and 11°N for oceanographic observation between April and May 2019. The *in-situ* observations show that the major eddy impacts are restricted to the upper 300 m of the water column and are traceable down to 1500 m water depth. In the eddy core at 80 m water depth an extreme positive temperature anomaly of +8°C, a negative salinity anomaly of -0.75 psu, a positive fluorescence anomaly of +0.8 $\mathrm{mg\,m^{-3}}$ and a positive dissolved oxygen concentration anomaly of 160 $\mu\,\mathrm{mol\,kg^{-1}}$ are observed. Considering the water volume trapped within the eddy, an average positive westward zonal heat transport anomaly of 85 $\times 10^{12}$ W and an average westward negative zonal salt transport of -2.2 $\times 10^6\,\mathrm{kg\,s^{-1}}$ are estimated. The heat transport is the equivalent of 1% of the total annual zonal eddy-induced heat transport at this latitude in the Pacific Ocean. Understanding the dynamics of mesoscale eddies in this region of the Pacific Ocean is especially important in the light of potential deep-sea mining activities that are being targeted on this area.

## 1 Introduction

Mesoscale eddies are ubiquitous features of the world's ocean, having a typical horizontal scale of about 100 km and a lifespan that can last between tens and hundreds of days (Shcherbina et al., 2010; Washburn et al., 1993; Chelton et al., 2011). These key features of the oceans do not only play a fundamental role in the general circulation of the upper ocean, but also contribute significantly to long-distance and vertical transport of mass, heat, salinity, oxygen, and nutrients within ocean basins (Sofianos





and Johns, 2007; Chen et al., 2011; Chaigneau et al., 2011; Chelton et al., 2011; Zhang et al., 2014; Stramma et al., 2014; Dong et al., 2014; Hill et al., 2015; Sun et al., 2019).

The northeastern tropical Pacific Ocean (NETP) is a region characterized by frequent mesoscale eddy genesis with complex flow regimes (Willett et al., 2006). This region is influenced by different climate phenomena ranging from seasonal spatiotemporal variability (i.e., gap wind bursts: Liang et al. (2009)) to large-scale events (i.e., El-Niño–southern Oscillation: Alexander

et al. (2012)). Recent economic interests in the extraction of deep-sea polymetallic nodules in the Clarion-Clipperton Zone (CCZ) has strongly motivated scientific expeditions and deep-sea environmental studies in this region. Nonetheless, very little attention has been paid to the study of eddies and their impacts on hydrography and the anomalous transport of heat and salt in this region. In the framework of the European MiningImpact 2 project (2018-2022), we have gained new insights into mesoscale eddies and their effects on local, deep-sea hydrodynamics and seawater properties in this region, by combining

different sets of data, including reanalysis data, *in-situ* observations and satellite altimeter data.

Due to recent advances in the development of EDA and the widespread availability of satellite altimeter data, our knowledge on eddies and their surface characteristics has improved significantly (e.g., Chelton et al., 1998, 2000; Chaigneau and Pizarro, 2005; Chaigneau et al., 2008; Liang et al., 2009). Applying an EDA on 24 years of altimeter data (Purkiani et al., 2020), it has been shown that between 4 and 6 long-lived (lifetime > 90 d) anticyclonic eddies (ACEs) are generated in the NETP annually,

mostly initiated off the west coast of Mexico by the Tehuantepec (TT) and Papagayo (PP) gap winds, which can travel distances greater than 1000 kilometers westwards into the open ocean. The analysis of these long-lived eddies shows that they spread in a narrow meridional corridor between 10° and 12°N in the interior of the ocean taking 5-6 months to reach the designated region for future deep-sea mining in the eastern CCZ. Aleynik et al. (2017) have shown that deep-reaching eddies may have the potential to influence the transport of deep-sea sediment plumes produced by a potential polymetallic nodule mining activity

in this region.

The different sea surface height anomalies induced by ACE (positive) and cyclonic eddies (CE, negative) are associated with significant geostrophic velocity anomalies. Furthermore, eddies can carry distinct temperature and salinity anomaly ($T'/S'$) structures in the subsurface that can be identified by hydrographic measurements (Chelton et al., 2011; Stramma et al., 2014; Czeschel et al., 2018). Due to different vertical movements of the thermo/pycnocline inside eddies, relatively warm-fresh and

cold-salty water masses are generally present in the cores of ACEs and CEs respectively (McGillicuddy Jr, 2014).

Research on mesoscale eddies has steadily increased in recent years with a lot of observations focusing on the South China Sea (SCS). An extremely cold, cyclonic eddy and its significant impact on the vertical heterogeneity of seawater in the SCS was reported by Hu et al. (2011). The eddy core at 50 m water depth was shown to reveal a temperature anomaly of up to -8°C compared the rim of the eddy, with a maximum vertical stretching of 250 m into the deeper layers and a vertically

tilted central axis. Using a series of satellite images before and after the impact of a cold-core cyclonic eddy in the SCS, Hu et al. (2011) suggest that the concentration of biogeochemical properties of seawater i.e., nutrient budget, primary production, phytoplankton assemblage at the Luzon Strait in a relatively unproductive season was even higher than during the winter time due to higher eddy activity. Later, long-term Argo profile data from the SCS, revealed that ACEs have a warmer (+1.4 °C) and fresher (-0.16 psu) eddy core that is located at 90 m water depth (He et al., 2018). Conversely, CEs showed a colder (-1.5 °C)





and more saline (+0.15 psu) core. While at both cases in their study the temperature anomaly T′ reaches down to a depth of
       400 m, the salinity anomaly S′ is limited to the upper 150 m.

         Using spatially high-resolution *in-situ* measurements from the northern SCS, Nan et al. (2017) identified a large subsurface
       anticyclonic eddy with a horizontal diameter of 470 km. Despite the large size of the eddy, the temperature anomalies of the
       eddy core located at 450 m depth did not exceed more than +3.5°C. However, the T′ induced by this subsurface eddy at 1000
m depth did reveal very weak yet discernible geostrophic velocities in the deeper layers, reflecting the clockwise rotation of an
       ACE. In contrast to previous observations, this eddy was characterized by a positive S′ of 0.25 psu trapped at 450 m depth with
       a complex vertical structure of multiple eddy cores at different depths. A typical ACE with positive T′ (0.65 °C) and negative
       S′ (-0.02 psu) in the eddy core was reported for three long-lived eddies in the northern SCS (Nan et al., 2011). Although the
       core of the eddy with maximum anomalies was located at a rather shallow depth of 65 m and the hydrographic impacts on
seawater properties were relatively weak, the anomalies in the current velocities extended to much greater depths (900 m).
       These analyses show that eddies with larger sea surface height anomalies (SSHA) and weaker vorticities extend deeper than
       eddies with smaller SSHA and stronger vorticities.

         The presence of eddies and their effects on the formation of seawater anomalies have also been addressed in other ocean
       basins. A historical record of Argo floats and satellite altimeter data in the Peru basin reveals the key differences in the
vertical structure of mesoscale eddies of different types (Chaigneau et al., 2011). This study shows that the core of CEs are
       centered at 150 m depth, i.e. much shallower than the average core of ACEs located at 400 m depth. The reason for different
       vertical extensions is attributed to the mechanisms involved in eddy formation. While instabilities of the surface equator-ward
       coastal current are the main source for the development of CEs, the ACEs are shed from the subsurface pole-ward Peru-Chile
       undercurrent. Despite the differences observed in the vertical extension of mesoscale eddies and their formation mechanisms,
the maximum T′ ($\pm$ 1 °C) and S′ ($\pm$ 0.1 psu) as well as the typical volume anomaly fluxes transported by eddies ($\sim$ 0.1
       Sv) for both types are of the same magnitude. Moreover, these eddies transport significant heat and salt anomalies ($-3 \times 10^{11}$
       W and $-8 \times 10^3$ kg s$^{-1}$) into the open ocean in this region. The vertical structure of mesoscale eddies and their impacts on
       anomalous transport of heat, salt and oxygen off the Chilean coast were investigated using a series of long-term moorings at
       the Peruvian-Chilean upwelling system (Colbo and Weller, 2007; Stramma et al., 2014; Czeschel et al., 2018). An extreme
anomalous water mass with higher temperature and lower salinity was observed over the entire depth of moorings from ocean
       surface to 450 m water depth (Stramma et al., 2014). The isolated water mass and the shoaling of the seasonal pycnocline in
       the ACE could resulted in high productivity and an over-saturation of oxygen in the near-surface layer. In contrast, a reduction
       to almost zero oxygen in the subsurface layer of the eddy core (145-450 m) was observed, due to remineralization of organic
       material by bacteria and zooplankton during the 11 months travel time. In contrast to above mentioned studies that did not
show strong positive seawater temperature anomalies, a good illustration is shown in the SCS (Chu et al., 2014). A long-lived
       ACE observed in the SCS, with diameter of 400 km, had a temperature anomaly of +7.7 °C that extended vertically to 500
       m. From the southwestern Canada Basin, (Nishino et al., 2011) describe an unusually large warm-core ACE of $\sim$ 100 km in
       diameter, 4-5 times larger than the average eddy size at Arctic latitudes), which was up to 5 °C warmer than its surrounding
       water..





Apart from local impacts of eddies on T and S anomalies, Sun et al. (2019) show that between 1/3 and 1/2 of the global heat transport by the ocean in mid-latitudes is associated with eddy heat transport that is confined to the upper 1000 m of the ocean. It has also been shown that eddies in the SCS generate meridional heat and salt transports that are confined to the upper 100 m and 300 m of the water column, respectively, and cause a transport equal to 30% of the annual mean water transport in the Luzon strait (Hu et al., 2011).

Despite the substantial mesoscale eddy activity in the NETP, recent advances in measurement techniques of ocean eddies, and the increasing likelihood that deep-sea mining for polymetallic nodules will materialise in the near future in this region, no previous studies have addressed the eddy impacts on seawater and the anomalous transport of heat and salt in this region. The vertical extension of eddies, their impacts on the deep-sea environment, and the time lag for the transfer of eddy effects from the sea surface to the deep-sea, are potentially of major interest for assessing the cumulative effects that changes in water

column hydrography can potentially have on mining-related impacts such as the dispersal of a sediment plume. Combining different datasets obtained from a measurement campaign in the CCZ in 2019, we provide a deeper insights into eddy impacts in this region.

## 2    Data and methods

*In-situ* hydrographic observations assist scientists to study eddies and their effects on subsurface temperature and salinity

anomalies in the ocean. There is a scarcity of available *in-situ* measurements in the open ocean of the NETP. As a part of the European MiningImpact project (Ecological Aspects of Deep-Sea mining), an ACE was analyzed in the eastern CCZ during the expedition SO268 with RV SONNE, from 01 April to 17 May 2019 (Haeckel and Linke, 2021).

### 2.1    CTD observations

A total of seven Sea-Bird SBE9 plus conductivity-temperature-depth (CTD) measurements were taken throughout the entire

water column over a period of five weeks from 02 April 2019 to 17 May 2019 (Table 1). Depending on the position of the eddy as obtained from altimeter data analysis, various CTD casts were conducted at the center and rim of the eddy to characterize its seawater mass properties. The CTD casts 08, 09, 10, and 11 were taken within 2 days along a zonal transect in E-W direction across the nearly 2 degrees of longitude influenced by the eddy. CTD 11 was obtained outside of the eddy periphery as a reference station in order to determine the eddy impacts on the seawater properties relative to the eddy (Table 1). After that,

CTD casts were performed at the same location of CTD 11 (CTDs 15, 16 and 17) within 5 weeks at the western rim, center and eastern rim of the eddy while it was moving towards the polymetallic nodule exploration contract areas of the CCZ. The CTD frame was also equipped with an upward-looking and a downward-looking Nortek Aquadopp 2 MHz recording every second with 20 bins with 0.5 m blanketing. During the upcast, the CTD paused for 10 minutes at 50, 150, 250 m and every 500 m below it to collect current data that was not distorted by movement of the CTD. The data quality from the upward-looking

Aquadopp was poor; therefore, we only show data from the downward-looking Aquadopp.





**Table 1.** Positions and characteristics of CTD and ADCP deployment across the eddy

| Station/transect | Lat(°N) | Lon(°W) | Deployment time (year days, UTC) | Local depth (m) | Eddy presence | Marker |
|---|---|---|---|---|---|---|
| CTD 08 | 11.749 | -113.116 | 04/02 14:30-00:00 | 4144 | yes, center | ▢ |
| CTD 09 | 11.766 | -114.036 | 04/03 05:00-11:00 | 4111 | yes, west rim | ▢ |
| CTD 10 | 11.799 | -114.94 | 04/03 16:00 to 04/04 02:00 | 4122 | yes, west rim | ▢ |
| CTD 11 | 11.859 | -117.013 | 04/04 12:30-18:30 | 4200 | no | ▢ |
| CTD 15 | 11.859 | -117.014 | 04/27 05:30-15:00 | 4200 | yes, west rim | |
| CTD 16 | 11.86 | -117.013 | 05/11 03:00-09:30 | 4189 | yes, center | |
| CTD 17 | 11.86 | -117.013 | 05/16 12:30-18:30 | 4213 | yes, east rim | |
| A1-A2 | 13.806 to 11.75 | -111.841 to -113.116 | 04/01 17:05 to 04/02 07:30 | - | - | |
| A2-A3 | 11.75 to 11.863 | -113.116 to -117.011 | 04/02 07:30 to 04/06 18:15 | - | - | |

## 2.2 ADCP measurements

While the ship was underway and when being on stations, the hull-mounted Acoustic Doppler Current Profiler (ADCP 75 kHz), almost continuously measured the upper-ocean current structure. To calculate the correct ocean currents from a moving vessel, the exact speed vector of the vessel, which is derived from the navigation and motion sensors, were subtracted from the flow fields observed by the ADCP. The vessel-mounted ADCP covered 40 bins of 16 m from 24 m to 648 m water depth. Quality control analysis of the measurements shows that the ADCP recorded meaningful data for a depth range of up to 500 m. The location of transects and detailed information on positions and deployment time are shown in Figure 1 and Table 1.

## 2.3 Satellite data and eddy detection process

The altimeter data used in this study to identify and track mesoscale eddies were obtained from the COPERNICUS Marine Environment Monitoring Service (CMEMS), and are freely available at http://marine.copernicus.eu. The altimeter data used in our study are the near-real-time level-4 sea surface height and derived variables measured by multi-satellite altimeter observations over the global ocean with 1/4° spatial resolution and a daily temporal resolution from 01 Jan 2018 to 01 June 2019. The position of the eddy was tracked in near real time using an EDA at the University of Bremen in Germany and subsequentlly communicated to the scientists on-board RV SONNE to define the sampling programme and locations (Figure 2). In order to perform the most efficient CTD observations, an EDA and a simple forecasting method were simultaneously used to determine the center, shape, radius and translation velocity on a daily basis. Among the identified long-lived eddies, the one which was anticipated to most likely pass through the location of the German exploration contract area (eastern CCZ) was selected. A previous study in this area (Purkiani et al., 2020) shows that most of the long-lived ACEs in the NETP converge to a meridional corridor between 10° and 12°N. In that study, an automated EDA developed by Nencioli et al. (2010) was applied and tested on the long-term sea level anomaly (SLA) data to identify and characterize the mesoscale eddies in this region. To forecast the eddy center positions a simple prediction method based on the average eddy position within the 7 days prior to the desired day




was performed as follows:

$$[X,Y]_t = [X,Y]_{t-1} + T \times [\overline{U},\overline{V}] \tag{1}$$

where $X_t$ and $Y_t$ are the new geographical position, $X_{t-1}$ and $Y_{t-1}$ are the old position, $\overline{U}$ and $\overline{V}$ are the average translation
velocity of the eddy $(\mathrm{cm\,s}^{-1})$ based on the 7 days prior to time $t$ and T is the time interval in second.

### 2.4 Reanalysis products

Since the 3D structure of the eddy cannot be inferred from individual CTD casts, an eddy-resolving global ocean reanalysis
product was employed (Drévillion et al., 2018), with a horizontal resolution of 1/12° and 50 vertical layers covering the time
period between January 2018 and June 2019. The aim was to distinguish $T'$ and $S'$ induced by eddies by comparing the seawater
properties in the presence of an eddy with one year of climatological T and S data. Seawater properties were interpolated to
the position and time of the CTD casts, validated against them and then presented in our results.

### 2.5 Heat and salt transport equations

Following previous studies (Dong et al., 2014; Yang et al., 2015; Lin et al., 2019), we obtained an estimate of the zonal heat/salt
transport fluxes induced by eddies in this region. The composite three-dimensional structures of temperature anomalies, salinity
anomalies, and swirl velocity were considered and the zonal heat $(H_f^x, \mathrm{W\,m}^{-2})$ and salt $(S_f^x, \mathrm{kg\,s}^{-1}\,\mathrm{m}^{-2})$ fluxes calculated as
sollows

$$H_f^x = \rho C_p U' T', \tag{2}$$
$$S_f^x = 0.001 \rho U' S', \tag{3}$$

where $\rho$ is the density of sea water, $U'$ is the zonal velocity deviation from mean long-term velocity and $C_p = 4187 \mathrm{J\,kg}^{-1}\,{}^{\circ}\mathrm{C}^{-1}$
is the specific heat capacity. The factor 0.001 converts salinity to a salinity fraction (kg of salt per kg of seawater). The transports
estimated above are usually zonal, since eddies in the NETP often travel westward. The meridional components can be readily
obtained by replacing $V'$ and the proper $T'$ and $S'$ in the zonal transects. As a result the heat (HT) and salt (ST) anomalies
transported by an eddy can be estimated by integrating the eddy's area from the trapping depth $(h_t)$ to the surface :

$$HT = \int_{h_t}^{0} \int \rho C_p T' U' \, dydz, \tag{4}$$

$$ST = \int_{h_t}^{0} \int 0.001 \rho S' U' \, dydz, \tag{5}$$

In contrast to assuming a depth of no motion at 1200 m (e.g. He et al. (2018)), heat and salt transport were calculated by
integrating equations 4 and 5 for the entire water depth.



## 3 Results

### 3.1 Eddy formation and its characteristics

Using an EDA on 24 years of altimeter data, Purkiani et al. (2020) showed that between 4 to 6 long-lived (lifetime > 90 d) ACEs are generated annually in this region, mostly in the vicinity of the TT and PP gap winds. They travel distances of more than 1000 kilometers into the open ocean through a narrow meridional corridor between 10° and 12°N and reach the eastern CCZ (German contract area) with a time delay of approximately 5-6 months (Figure 1a).

The SLA data from 01 January 2018 to 01 July 2019 were used to identify mesoscale eddies in the NETP. A total of 763
CEs and 745 ACEs with a lifetime of more than 7 days were identified, of which only 2 CEs and 7 ACEs were characterized as long-lived eddies (life-time> 90 d). We focused our efforts on one particular ACE that originated at TT and that crossed over the German contract area in the CCZ at the time of the SO268 expedition. The analysis of wind field data in the TT gap wind region showed a sudden development of a strong wind burst event exceeding $10 \, \mathrm{ms}^{-1}$ for a short period of 9 days between 7 and 16 July 2018 (not shown here). The statistical characteristics of the long-term wind data over a 31-yr period at the isthmus
of TT indicate a strong seasonal variability, with maximum values in the boreal winter (November-February) and a relative maximum in July (Romero-Centeno et al., 2003). Analysis of altimeter data shows that the secondary maximum wind intensity in the TT Gulf formed a long-lived ACE in this region between 15 and 23 July 2018. The formation of a long-lived ACE in July is unusual and has not been reported in previous studies from this region. The eddy remained stationary in the Gulf of TT from August to December 2018 and only separated from the coast in January 2019. The movement of the eddy center into the NETP
for the next 9 months is depicted by the blue circles in Figure 3a (time interval of 7 days). The mean diameter of the eddy is estimated at 230 km with an average translation speed of $11 \, \mathrm{cms}^{-1}$ since January 2019. Our series of *in-situ* observations to elucidate eddy impacts on seawater characteristics occurred at a distance of 2050-2400 km away from origin (between 02 April and 16 May 2019).

### 3.2 Water mass properties in the NETP

The NETP is influenced by two large subtropical gyres in its upper layers: the North Equatorial Current (NEC) to the north of the eddy belt and the North Equatorial counter-current (NECC) to the south of the eddy belt. The annual mean temperature and salinity profiles in 2018 obtained from the reanalysis model and averaged over an area of 1° by 1° centered at the position of CTD 08 are shown in Figure 4a-b. The shaded area for T and S shows the temporal standard deviation at each depth. Four prominent water masses characterize the water column in this area: Tropical Surface Water (TSW), Equatorial Surface
Water, Subtropical Surface Water and California Current Water (Fiedler and Talley, 2006; Portela et al., 2016). Here, TSW is characterized by a sea surface temperature reaching up to 28.5°C and a sea surface salinity of about 33.7 psu. Beneath the warm and low-salinity TSW, a pronounced pycnocline caused mainly by a strong, shallow thermocline and reinforced by halocline is observed, that does not, however, contain a distinct water mass of any substantial volume (Wyrtki, 1996; Fiedler and Talley, 2006). Below the pycnocline, a water mass with S>34.6 psu and T between 10°C and 14°C characterizing the
Eastern North Pacific Central Water (ENPCW) is evident. Between 500 m and 1000 m, the T-S diagrams (Figure 4b) show





a mixture of Antarctic Intermediate Water (AAIW; S>34.5 psu, 2°-10°C) and Pacific Intermediate Water (PIW; 34.5<S<34.9 psu, 4°-9°C). In deep waters >2500 m, North Pacific Deep Water (1.2-2°C; S>34 psu) is present. Below 4000 m depth a water mass with temperature of about 1.2°C and salinity of 34.68-34.7 psu is attributed to Lower Circumpolar Water (LCPW), which in the eastern Pacific Ocean is characterized by high oxygen concentrations (Fiedler and Talley, 2006). The surface water layer

is saturated with dissolved oxygen (>180 $\mu$mol kg$^{-1}$). The oxygen concentration decreases rapidly to sub-oxic concentration (<10 $\mu$mol kg$^{-1}$) below the oxycline at 130 m water depth. Oxygen-poor water is found below the oxycline, which extends down to 900 m depth. This oxygen minimum zone with a thickness of at least 600 m is evident in the entire NETP due to a lack of oxygen-rich deep water formation in the Pacific Ocean (Kamykowski and Zentara, 1990; Karstensen et al., 2008; Stramma et al., 2014). The T-S diagram of the no-eddy situation (CTD 11, orange line in Figure 4b) remains in the range of the standard

deviation as shown by the shaded areas in the original T-S profiles (Figure 4a ). However, the T-S diagrams of the seawater in the presence of the eddy deviates from the annual standard deviation at the sea surface and at depths between 300 and 550 m (Figure 4a-b).

### 3.3   Temperature, salinity, and oxygen anomalies induced by the eddy

The vertical temperature (T), salinity (S), and dissolved oxygen (DO) profiles obtained from an E-W transect across the eddy

center (Figure 1b) measured within 2 days from 02 April to 04 April 2019 are shown in Figure 5a-c-e. To further illustrate eddy impacts, we have determined the anomalies of these seawater properties, which are considered as deviation from the background ocean state, by subtracting a no-eddy state from any individual vertical hydrographic profile taken from the eddy region (Figure 5b-d-f). The eddy has significant but variable impact on the T, S and DO profiles of the water column. In the presence of the eddy, all CTD profiles show warmer, fresher, and more oxygen-rich seawater in the upper 500 m (Figure 5).

The major impact of the eddy on seawater T is confined to the upper 500 m of the water column, with a maximum anomaly of +7.8°C at 87 m water depth, which is the position of the seasonal thermocline in this region. The seawater T anomalies decrease to intermediate water depth, with only a weak positive anomaly of 0.05°C at 1500 m water depth, and become negligible below that. A maximum S anomaly of -0.74 psu is located at 82 m water depth. In contrast to T, S shows no significant alteration below 500 m. However, the second (positive) and third (negative) anomalies known as the weak salinity dipoles (Nan et al.,

2011) were only discernible at 150-300 m and 300-700 m depth. The DO distribution is also affected by the eddy, with an increase up to 130 $\mu$mol kg$^{-1}$ in the subsurface layers. However, the eddy appeared to negatively affect the oxygen-poor water content below the pycnocline and increases the hypoxia at depths between 150 and 500 m. A positive significant DO anomaly (+20 $\mu$mol kg$^{-1}$) is observed in the deeper part of the ocean between 800 and 1500 m. In contrast to other seawater properties, the DO anomalies are more pronounced throughout the entire water column, even near the seafloor where positive anomalies

of 1-4 $\mu$mol kg$^{-1}$ were still being registered. Interestingly, the intensity of anomalies observed in T, S, and DO is larger at the two CTD positions close to the eddy center (east and west of the eddy center) compared to the western rim of the eddy (compare the black and blue lines with the red line in Figure 5 b-d-f). According to the criteria of Montégut et al. (2004) for the calculation of the mixed layer depth (MLD), the eddy has deepened the MLD by 8 (12) m from 66 m in a no-eddy situation to 74 (78) m at the western (eastern) segment of the eddy. Further analysis shows that the thermocline is located at 71 m during a




no-eddy situation. The ACE deepened the thermocline layer by 13 to 24 m from 72 to 95 m water depth and thickened it by 10 m from 50 m to 60 m at its eastern and western segments. The deepening of the thermocline is in agreement with the classical impact of anticyclonic eddies on seawater characteristics derived from both local (Chen et al., 2010) and global measurements in the oceans (Gaube et al., 2019).

The impact of the eddy on seawater properties was further investigated by conducting repeated CTD measurements while
the eddy was approaching and passing over the location of the CTD 11 station within 5 weeks between 04 April and 16 May 2019 (Figure 1b). A larger maximum T anomaly of +8.8°C located slightly deeper (at 90 m water depth) was observed. The maximum S anomaly reduced to -0.68 psu in the eddy core at 88 m water depth. The DO showed higher anomalies of +162 $\mu$mol kg$^{-1}$ at 86 m water depth. In agreement with the measurements taken across the eddy, T anomalies could be distinguished down to greater depth than S anomalies. Also, the observed anomalies of all seawater properties were found to be stronger in
the eastern and central segment of the eddy compared to the western segment, and the major impacts were confined to the upper 500 m of the water column. The T anomalies observed in our study are significantly larger than in most previously reported measurements, in which a warmer eddy core of between 1.5° and 5°C was registered e.g. Indian Ocean (Lin et al., 2019; Yang et al., 2015), in the Pacific Ocean (Ji et al., 2018; Chaigneau et al., 2011; Stramma et al., 2014; Czeschel et al., 2018), in the SCS (Nan et al., 2017; He et al., 2018; Chen et al., 2010), and in the Arctic Ocean (Nishino et al., 2011). However, one similar
anomaly induced by an extremely large ACE with a warm (+7.7°C) and fresh core (-0.8 psu) was observed in the SCS by Chu et al. (2014).

To illustrate the seawater anomalies induced by the ACE in the water column from 02 April to 04 April 2019, the lateral E-W transect (A2-A3, CTDs 08-09-10) through the eddy center is shown in Figure 6. The T anomaly reaches a maximal of up to +8°C in 90 m water depth near the eddy center, whereas anomaly values of +0.5°C and +0.05°C extend down to 350 and
1500 m water depth, respectively. This is accompanied by a decrease in the salinity of seawater by -0.7 psu in the eddy core. Following the T and S profiles, the seawater density decreases by 2.5 kg m$^{-3}$ in the eddy core. The oxygen anomaly displays a more complex pattern, expressed most clearly at the eddy core with maximum positive anomaly value of +130 $\mu$mol kg$^{-1}$ but negative values in the sub-surface regions from 300 to 1000 m water depth. The positive oxygen anomalies reach to the seafloor with a significant impact ($> 20 \mu$mol kg$^{-1}$) distributed below mid-water depth of 1000 m. The impact of the eddy on
the fluorescence (FL) profile is mainly reflected by a deepening of the maximum fluorescence in the eddy center (1.45 mg m$^{-3}$) of about 15 m, which, in comparison to a no-eddy situation, causes a maximum anomaly of +0.6 mg m$^{-3}$ at the eddy core (Figure 6e). Our analysis indicates that no differences in FL are observed in the eastern and western segments of the eddy. Therefore, a more symmetric pattern of FL anomalies emerges in the vertical profiles.

### 3.4  ADCP observations across the eddy

The current velocity measured by the ship-borne ADCP during the transects A1-A2 (northeastern segment to the eddy center) and A2-A3 (eddy center to the western eddy rim), is shown in Figure 7. The current velocity at the surface (40 m) is greater than in 200 m with maximum velocities of 40 and 20 cm s$^{-1}$, respectively. Moreover, current data obtained during the A1-A2 transect show a consistent clockwise rotation as is typical for an ACE. Moving along the A1-A2 transect, current velocities





increased at the northeastern rim of the eddy and decreased towards the eddy center. The station carried out at CTD 08 revealed

a multi-directional current at the eddy center, with a lower magnitude compared to the transect. The current velocities during

the A2-A3 transect are generally lower, probably due to the incoherent SLA with lower magnitudes in the western part of the

eddy. However, the northward current directions confirm the typical rotation of an ACE while it moved westward (Figure 7).

Moving from the eddy center to the western edge of the eddy along transect A2-A3, the current velocity increased to swirl

velocities larger than $45 \ \mathrm{cms}^{-1}$ which satisfies the eddy nonlinearity criteria ($U/c > 3$, where U is swirl and c is translation

velocity of an eddy) defined by Chelton et al. (2011). The advective nonlinearity parameter $U/c$ is particularly useful for char-

acterizing mesoscale eddies, since a value of $U/c > 1$ implies that an eddy can transport heat, salt and potential vorticity, as

well as biogeochemical properties such as nutrients and phytoplankton. Nonlinear eddies can thus have important effects on

heat flux and marine ecosystem dynamics (Chaigneau et al., 2008; Hu et al., 2011; Gaube et al., 2014).


### 3.5    3D structure of the eddy

In the absence of detailed spatial and temporal observations, we have used reanalysis products to explore the three-dimensional

structure of the eddy. A water column with a size of 1° by 1° centered at the eddy core on 03 April 2019 is considered, and

ocean temperature, salinity and background current velocities are shown at 80, 700, 1500, and 3000 m depths (Figure 8). The

low vertical resolution of this product (500 m) in the deep ocean may not be sufficient to depict a realistic eddy shape at those

depths, but our reanalysis-observation comparison shows a reliable correlation from the surface down to 3000 m water depth.

The geostrophic velocity reaches its maximum of $50 \ \mathrm{cms}^{-1}$ at 80 m water depth. The seawater T anomaly isolated in the eddy

shows a more symmetrical pattern than what we observe in the S field, and it is visible at 700 and 1500 m water depth with a

significant current pattern describing the specific rotational feature of an ACE at this depth (Figure 8a). A current intensification

in deeper parts of the ocean at 3000 m is evident as well as a negative S anomaly above 400 m and a positive anomaly at a

depth of 1500 m (Figure 8)

We follow the eddy while it is moving westward and show the 2D structure of vertical T, S, $\rho$ and velocity anomaly profiles by

removing the corresponding climatological profiles from the relevant profiles at the corresponding time and transects (Figure

10a-h). The eddy core is located at 80-90 m and does not shoal while the eddy evolves in the ocean. The eddy footprint is more

clearly distinguishable from the T anomaly than from the S anomaly and remains dominant while the eddy passes through the

area. The magnitude of T′ and S′ reduces to values below 8°C and -0.3 psu, respectively in the beginning of May and tends

to gradually reduce by mid May. The T anomaly decreases significantly below 300 m, although the impact of the eddy on T

in intermediate water is not yet negligible. Consistent with our *in-situ* observation, the analysis of the dynamic height of 5°C

isotherms shows a vertical descent of the isotherm by about 200 m, from 910 m during the no-eddy situation to 1100 m during

eddy passage. The vertical extension of the eddy as reflected by the T anomaly can be seen down to a depth of ~1500 m. The

seawater anomalies obtained from reanalysis products produce a bottom-up, straight conic shape for the eddy with a height of

about 1500 m and a radius of 100 km at the ocean surface, and with a tilted tail in the deeper ocean (Figure 11). On average, the





volume of trapped fluid transported by the eddy based on this cone shape is estimated to be $10\times10^{12}\,\mathrm{m}^3$. This is of a similar magnitude compared to previous estimations in the South Pacific Ocean (Chaigneau et al., 2011). The center of the S anomaly

observed at different depth levels coincides with that of the T anomaly, with a slightly elevated height from 30-150 m. They cannot be observed in the deeper layers. Interestingly, the positive S anomaly is observed in all profiles from the ocean surface to 35 m depth (Figure 10e-h). The eddy core with maximum negative anomaly of about -0.5 psu is located at 75 m water depth. In agreement with the measurements, the S anomalies gradually weaken, so that the maximum of S anomalies reduced to -0.3 psu on 14 May. The vertical extents of the S anomalies are much lower than those of the T anomalies, and disappear below 300

m. The limited depth of S anomaly have been reported earlier in the SCS, where a barrier salinity layer was found to prevent anomalies from penetrating deeper into the ocean (Nan et al., 2016; He et al., 2018).

Tracking the passage of the eddy through time indicates that negative density anomalies are found inside the eddy, with a maximum anomaly of -2.5 $\mathrm{kg\,m^{-3}}$ at a depth of 80 m (Figure 10i-l). The pattern of density anomalies mainly follows the T structure, with extended impact in the deeper part driven only by temperature. In the upper layers, the zonal current velocity

anomalies show a clear anticyclonic pattern with an eastward direction on the northern segment and a westward direction on the southern segment of the eddy core (Figure 10m-p). The deeper part of the ocean (>1500 m) shows no significant anomalies, but this is likely due to the fact that the vertical resolution of reanalysis products is not good enough for this purpose.

The impact of the eddy on ocean flow field is further investigated with analyzing current velocities at different depths in an E-W transect as the eddy was moving towards the German contract area. The current velocity measurements obtained from

Aquadopps at the CTD (15, 16 and 17) stations have been plotted at different levels of 50, 150, 250, 500, 1000 and 1500 m water depth (Figure 9). The current velocities at the western rim of the eddy (CTD 15) generally show north-northeastward current directions in the upper layer. Below 500 m water depth variable currents are evident (Figure 9a). The current velocities at the centre of the eddy (CTD 16) show a clear southward current direction in the upper 1000 m water depth. The current velocities at the eastern rim of the eddy (CTD 17) also show a clear southward current direction in the upper 1000 m water

depth. Below that the ocean currents does not show a prevailing direction (Figure 9c). The current velocity observations at the E-W transect portrait the clockwise rotation of an ACE. Similar to the results of reanalysis products, no significant effects on ocean currents were observed at depths greater than 1000 m in our data.

### 3.6 Eddy induced heat and salt transport

Mesoscale eddies transport T/S anomalies owing to the advective trapping of interior water parcels, thus contributing to the

regional and global heat and salt transport in the oceans (Dong et al., 2014). It has been shown that the variability of eddy heat transport explains about 1/3 of the total heat transport in the oceans (Volkov et al., 2008; Sun et al., 2019). Therefore, it is interesting to investigate heat and salt transport induced by a typical long-lived eddy in the NETP. This issue can be addressed in an interdisciplinary context to further investigate the biological impact of an eddy on the deep ocean. The sign of the T/S anomaly tends to be opposite in cyclonic eddies with respect to anticyclonic eddies. In the eastern Pacific Ocean, with

differing CE and ACE movements and pycnocline displacement tendencies, but with both eddy types reflecting negative zonal velocity (westwards propagation), zonal heat and salt mass transport by the two types of eddies have opposite signs and thus



tend to create an equal balance. However, the greater amount of long-lived ACEs in the NETP may cause a dominant tendency toward a positive net heat flux in this region. Using Argo floats, it was shown that the annual average of zonally-integrated heat transport induced by eddies in the Pacific Ocean reaches up to -2 PW at 15°N, which can be attributed mainly to the CEs (Sun et al., 2019).

The ACE observed in our study is highly heterogeneous in both the horizontal and vertical direction. There is a considerable zonal current asymmetry across the eddy, with stronger westward currents in the southern part of the eddy and weaker eastward currents in the northern part of the eddy. The eastward segment observed in the northern part of the eddy appears to be more vertically stretched into the deeper layers of the ocean at all occasions (see Figure 10m-p). The rotational velocity in the eddy decreases with depth and becomes less than $0.1~\mathrm{ms}^{-1}$ below 1500 m depth. The vertical section of $T'$ and $S'$ shows that

there are asymmetric patterns in both profiles, with tilted isotherms and isopycnals in the northern part of the eddy (see Figure 10a-h). The combined asymmetries of T/S fields and current velocities observed in our study should demonstrate a net zonal heat (HT) and salt transport (ST) in this region.

Due to the asymmetries in seawater properties, the heat fluxes also show an asymmetric pattern with positive (eastward) and

negative (westward) values in the northern and southern sections of the eddy respectively, and with peaks of $\pm 1.5 \times 10^{7} \mathrm{W\,m}^{-2}$ at a depth of 90 m (Figure 12a-d). No significant features were found below 1000 m water depth. The meridional salt fluxes of the composite eddy have different signs compared to the heat fluxes and show positive (westward) and negative (eastward) salt transport in the southern and northern part of the eddy (Figure 12e-h). The salt transport at 80 m depth shows an absolute peak of $\pm 0.25 \mathrm{kg\,m}^{-2}\mathrm{s}^{-1}$. A common feature for the ACE is that the heat/salt fluxes were mainly concentrated above the

thermocline (<300 m)/halocline (<150 m). Similar patterns with slightly reduced magnitude of both heat and salt transport are observed at all occasions. The integrated HT and ST across the eddy area are strongly depth-dependent and mainly concentrated at the upper 500 m water depth. The HT is generally positive throughout the entire water column. It reaches its maximum of about +2 TW ($10^{12}\mathrm{W}$=TW) at 80 m water depth at 27 April and then drops off slightly (Figure 12i-l). The ST profile is slightly different throughout the water column with positive values in the top 50 m and negative values below that to the seafloor. The

maximum ST reaches up to $-2 \times 10^{4}~\mathrm{kg\,m}^{-1}\mathrm{s}^{-1}$ at 70 m water depth (Figure 12m-p). The vertical variation of HT and ST below 1000 m is almost negligible.

Integrated over the entire water column, HT varies between +26 and 150 TW, whereas ST variation is much smaller, ranging only between -1.5 and -2.8 $\times 10^{6}~\mathrm{kg\,s}^{-1}$. Compared to previous estimates that were carried out in the Indian Ocean (Yang et al., 2015; Lin et al., 2019) and in the South Pacific (Chaigneau et al., 2008), we find that the heat/salt transport advected in the

NETP is more effective and one order of magnitude stronger than in these other regions. In contrast, in the SCS eddy-induced heat/salt transport is similar to that observed in the NETP. The results underline the important role of eddies in the regional ocean circulation of the NETP.



## 4 Discussion

In the present study, the combined analysis of CTD profiles and a reanalysis dataset from the NETP provides new insights into
the seawater anomalies trapped inside an ACE while it evolves in the ocean. The analysis of the CTD profiles confirms the
classical picture of an anticyclonic eddy that is characterized by warmer and fresher water as compared to the ambient water,
which is in agreement with previously reported observations (Chaigneau et al., 2011; Nan et al., 2011; de Jong et al., 2014; He
et al., 2018). Most studies to date indicate a T anomaly in the range of 0.65°C to 5°C. The *in-situ* observations in our study
show an extremely warm eddy core with a T anomaly of about +8°C. Although such an extreme water T anomaly is not very
common in the ocean, a previous study of Chu et al. (2014) from SCS reported a long-lasting ACE with a T anomaly of 7.7°C.
The origin of the ACE observed in our study is in the TT Gulf, which belongs to the western Pacific warm pool region with
a sea surface temperature higher than 28.5°C in Spring (Wyrtki, 1996). Due to the high non-linearity of the eddy, the warm
sea surface water of this region is trapped in the eddy and transported far offshore (2500 km) into the ocean. In addition, the
unusually warm sea surface water could be associated with the positive phase of the ENSO cycle from late 2018 to early 2019
in this region (Cambronero-Solano et al., 2021). Moreover, the strong T anomalies may also stem from the size and intensity
of the sea level anomalies. He et al. (2018) found that an increase of 10 cm in SLA from 10 to 20 cm in the SCS corresponds
to an enhancement of the eddy core T anomaly of 1.5°C, from 2.5° to 4°C. Therefore, considering the relatively large SLA in
our study ($\sim$ 0.4 m), the unusually warm eddy core is consistent with the study of He et al. (2018) in the SCS.

Although the major eddy impact is restricted to the upper 300 m of the water column, the T anomalies are still visible at
1,500 m water depth, with depressed isotherms below the eddy core (e.g. 5°C isotherm shows a vertical displacement of 200
m at a water depth of 900 m). It has been shown that the vertical extent of eddies exceeds 800 m when the eddy amplitude is
greater than 20 cm (He et al., 2018). This is confirmed by the analysis of intense eddies in the SCS that extend vertically below
1000 m (Chu et al., 2014; Nan et al., 2017; Sun et al., 2018). Additionally, Zhang et al. (2016) show that a deep-penetrating
eddy can reach to 2000 m with a notably strong tilting of the eddy axis up to 100 km from the surface to the bottom. The tilted
eddy axis structure found in their study provides a good explanation for the previously observed 2-4 weeks time-lag between
surface eddies and their effects on near-bottom current intensification and alteration of predominant current direction in the
German exploration contract area in the eastern CCZ (Aleynik et al., 2017; Purkiani et al., 2020). Interestingly, the comparison
of the vertical extent of the eddy core in the same NETP area during different events in April 2013 (Aleynik et al., 2017) and
April 2015 (Purkiani et al., 2020) in the same region with the current study shows that despite of the profound impacts of the
former events, the effects of the eddy recorded in April 2019 do not reach down to the seafloor. This could be associated with
the larger SLA observed in 2013 ($\sim$ 0.58 m) and 2015 ($\sim$ 0.51 m) as compared to the lower SLA in 2019 ($\sim$ 0.4) in this region.

The core of maximum T and S anomalies centered at water depth of 80 m and 65 m, respectively, which is comparable to the
depth of the eddy core observed from ARGO profiles in the SCS (Chu et al., 2014; He et al., 2018). However, it is relatively
shallow compared to previously reported eddy core depth of 300 and 500 m in the northwestern and southeastern Pacific Ocean
(Chaigneau et al., 2011; Nan et al., 2017).



Finally, by analysing available global sediment trap data previously published by Lutz et al. (2007), a significant spatial distribution of Particulate Organic Carbon (POC) fluxes to the deep ocean and sedimentation rates in the NETP were reconstructed (Vanreusel et al., 2016; Volz et al., 2018). The spatial distribution of POC flux shows a lower flux of $1.3\ \mathrm{mg\,C_{org}m^{-2}d^{-1}}$ in low latitudes between 6° and 10°N. A little further to the north in the region between 10°N and 13°N, higher POC fluxes rang-

ing between 1.6 and $1.8\ \mathrm{mg\,C_{org}m^{-2}d^{-1}}$ were observed. Interestingly, the region of the deep sea characterized by higher POC fluxes closely corresponds to the region with a higher frequency of mesoscale eddy occurrence at the sea surface (Purkiani et al., 2020). Indeed, the temporal variation of vertical fluxes of biogenic, lithogenic and POC fluxes has been attributed to mesoscale eddy activity in the past (Buesseler et al., 2008; O'Brien et al., 2013).

## 5   Summary and conclusion

On the basis of altimetry data and *in-situ* hydrographic observations, the evolution and impact of a long-lived ACE on the seawater characteristics of the NETP was investigated during a research campaign that took place in the framework of the MiningImpact project to study the environmental impact and risks of deep-sea mining. The ACE originated in the Mexican Tehuantepec gap wind region during a strong wind event in summer 2018. Using an EDA, the eddy was successfully detected 9 months later at a distance of between 2,050 and 2,400 km off the Central American coast. This confirms that real-time altime-

ter data can be employed to detect eddies and precisely predict their path in the ocean. The particular ACE investigated here was characterized by a temperature anomaly of +8°C, a salinity anomaly of -0.75 psu, a dissolved oxygen anomaly of +160 $\mu\mathrm{mol\,kg^{-1}}$, and a fluorescence anomaly of $+0.8\ \mathrm{mg\,m^{-3}}$. The mixed layer depth in the ACE deepened by 29 m and thickened by approx. 10 m. The major eddy impact was confined to the upper 500 m of the water column, with the effect on seawater temperature being evident only until a water depth of 1,500 m. The eddy had no impact on the deeper part of the ocean. The

depth-integrated heat and salt transports induced by this mesoscale eddy reached to $\mathcal{O}$ (+10-100 TW) and $\mathcal{O}$ ($-10^{6}\,\mathrm{kg\,s^{-1}}$) respectively. This is comparable to 1% of the annual meridional heat transport at 10°N in the Pacific Ocean. The increased number of mesoscale eddies during ENSO (Palacios and Bograd, 2005) may cause an imbalance in the north Pacific heat and salt budgets. The results of this study will be useful for the deep-sea biological and biogeochemical community to investigate the connections between mesoscale eddies and deep-sea ecosystems. A combined oceanographic and biogeochemical numer-

ical simulation is recommended to study the potential role of mesoscale eddies on deep sea fertilizing and the growth rate of manganese nodules in the NETP. Whereas some mesoscale eddies passing over the area targeted for deep-sea mining change the near-bottom current regime at depths >4000 m and potentially may affect the deep seafloor and its ecosystems (Aleynik et al., 2017; Purkiani et al., 2020), this study has shown that this may not be the case for all mesoscale ACEs passing through the NETP. We suggest that the depth to which the eddies affect deeper water layers is related to the height of the sea level anoma-

lies observed at the surface. The potential role of eddy influence on the seafloor and its consequential effect on the altered dispersion of mining-related sediment plumes must be taken into consideration during impact assessments of future mining operations. Furthermore, we have shown that mesoscale eddies change the upper water column hydrographic and dynamic characteristics for several days to weeks as they pass over a particular location in the CCZ. Although the development and





testing of nodule mining technology is still in its infancy, most envisaged mining operations involve the discharge of ballast
waters, sediments and nodule debris from the surface production platform into intermediate waters below the oxygen minimum
zone, producing a discharge plume there. Eddies passing through the water column might change the background temperature
and salinity (density) conditions as well as current speeds and directions at such discharge points. This must also be taken into
consideration during future mining assessments and plume modelling activities.

## 6   Acknowledgement

This study was carried out in the framework of the European collaborative project MiningImpact and received funding through
the Joint Programming Initiative Healthy and Productive Seas and Oceans (JPI Oceans): German Ministry of Research and
Education (BMBF) grant no. 03F0812A-H. The altimeter products were produced by Ssalto/Duacs and freely distributed by
AVISO (www.aviso.oceanobs.com). The eddy resolving global ocean reanalysis products are available online at MERCATOR
GLORYS12V1 (www.mercator-ocean.fr). The authors confirm that there are no known conflicts of interest associated with this
publication. We thank the captain and crew members of RV SONNE for their assistance during the SO268 data campaign.

## 7   Author Contributions

KP and MH conceived the study. KP analyzed and interpreted the entire dataset, and wrote the first draft of the manuscript.
MH designed and conducted the field experiment. SH, HdS and KS contributed in CTD measurements. SH conducted the
Aquadopp measurements. PU and I-S.G were responsible for the collection of ship-mounted ADCP data. MW, AP and AV
contributed to the interpretation of the results. All authors contributed to the final draft of the manuscript.

## 8   Data availability

All CTD and Aquadopp data are under review to publish in PANGAEA. The doi will be provided later.

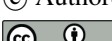



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



**Figure 1.** a) Northeastern tropical Pacific Ocean (NETP) and orographic features of Central America. The location of Tehuantepec (TT) and Papagayo (PP) gap winds are shown as red boxes. The general trajectory of long-lived eddies in the ocean generated at TT and PP are shown as red arrows obtained from the analysis of long-lived eddies obtained from Purkiani et al. (2020). The study area and location of hydrographic survey shown by the black box. b) Locations of CTD stations are indicated as red boxes. Two ship transects (A1-A2 and A2-A3; black lines) were carried out to cross the eddy and investigate surface currents using the ship's ADCP. Exact positions of CTD stations and surface currents on transects are described in Table 1. The bathymetry is shown by the blue background colors.



**Figure 2.** The concept of online communication between the research vessel in the NETP and the shore-based eddy detection group using online satellite data for determining the positions for hydrographic measurements. Figure is not to scale. Eddy, CTD and Aquadopp are shown.



**Figure 3.** (a) Satellite altimeter sea surface height (SSH) data for 27 April 2019. Eddies are detected using a geometry-based algorithm and their edges are shown by the solid black lines for every individual eddy center (black star). The background flow field is depicted by the black arrows with a space interval of four grid cells. The largest arrow corresponds to a surface current velocity of 1.5 ms$^{-1}$. The weekly eddy trajectory of the ACE analyzed in this study is shown by the blue circles from 01 Aug 2018 to 20 May 2019. Three configurations of the eddy are shown for (b) 03 April, (c) 11 May and (d) 16 May. The dashed black lines show the locations of meridional transects with maximum sea surface height at which vertical profiles of seawater properties were analyzed.



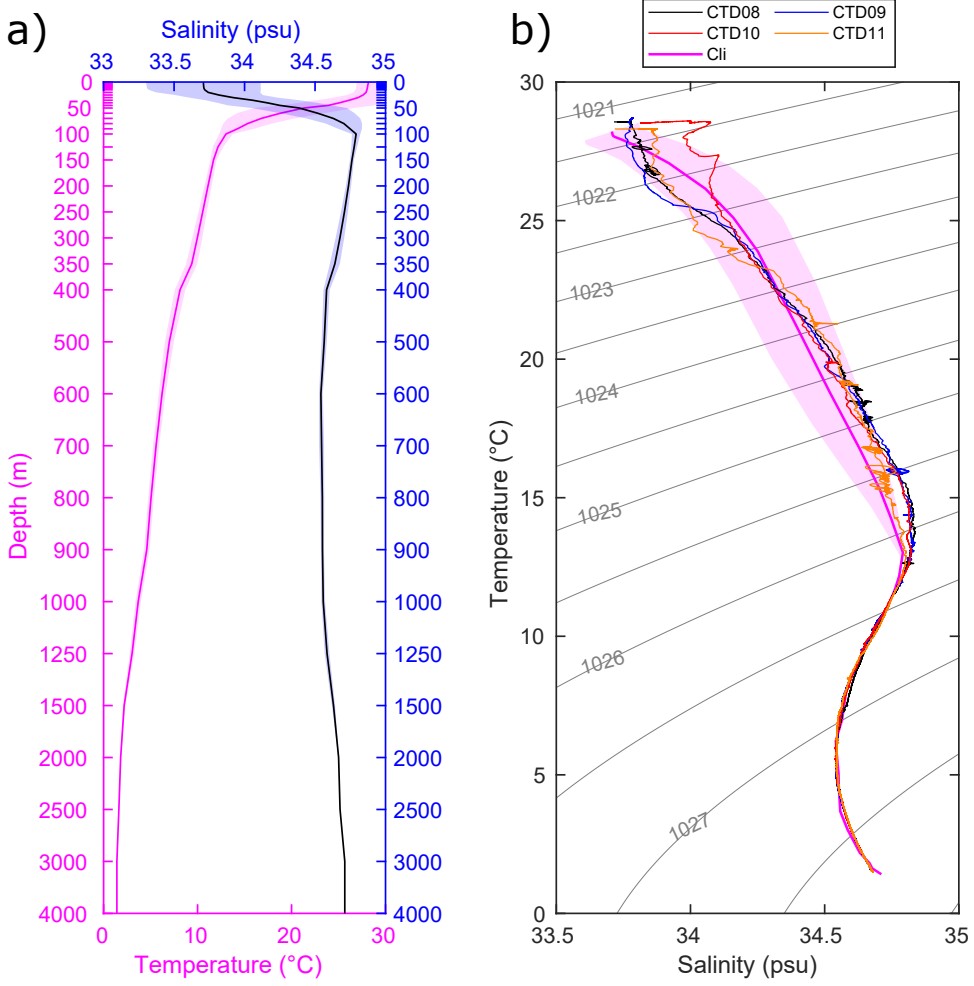

**Figure 4.** a) Annual mean temperature and salinity profiles in 2018 averaged over an area of 1° by 1° centered at the position of CTD 08 (see Table1). The temporal standard deviation is shown by the shaded area for each profile. b) T-S diagrams determined from climatology data (pink line) and the CTD casts listed in Table 1. Isopycnals are shown as grey lines with an interval of 0.5 kg m$^{-3}$

.

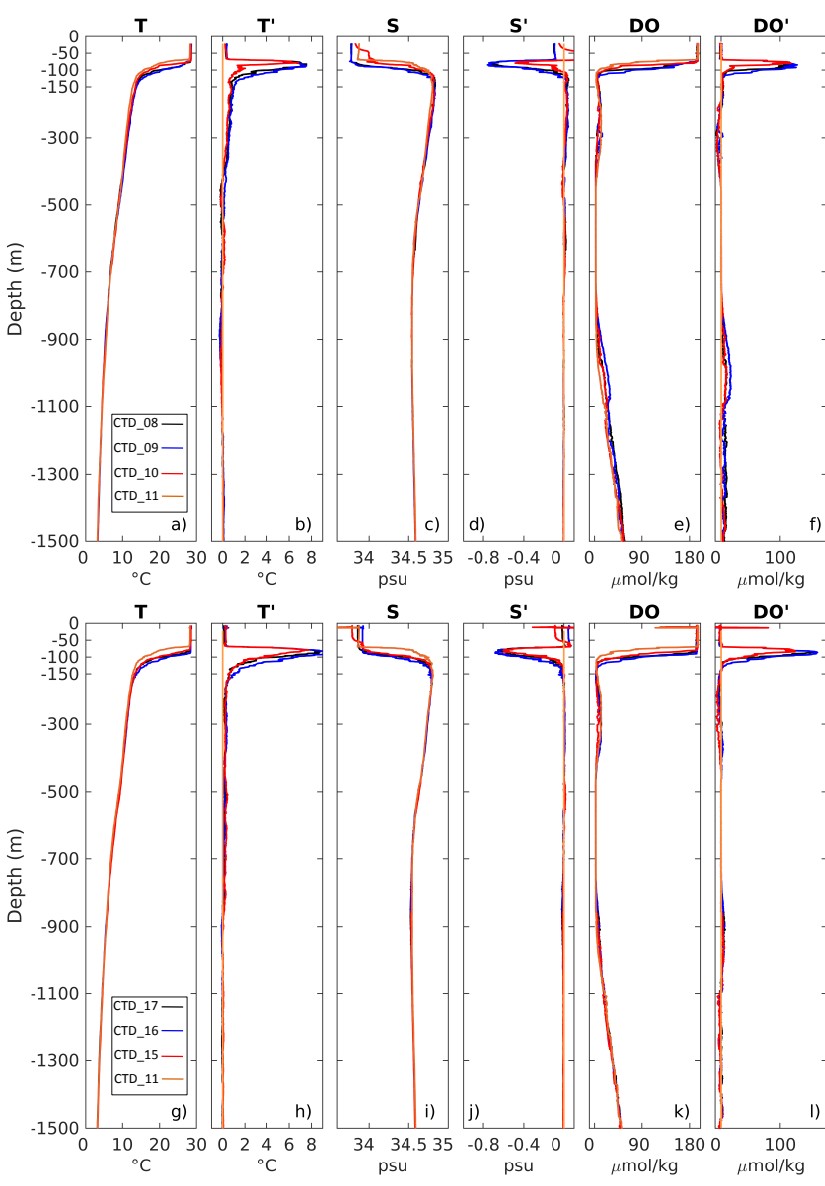

**Figure 5.** Vertical profiles of a) temperature, c) salinity, e) dissolved oxygen concentration and of anomalies in these respective parameters, from sea surface to 1500 m water depth for the E-W transect through the eddy center between 02 and 03 April. The black, blue, red, and orange lines show data from different CTD casts as shown in the legend. Subplots g), i), and k) show vertical profiles of temperature, salinity, and dissolved oxygen in replicate CTD casts at the same position between 04 April and 16 May. Associated anomalies of temperature, salinity, and dissolved oxygen are shown in h), j), and l) respectively.



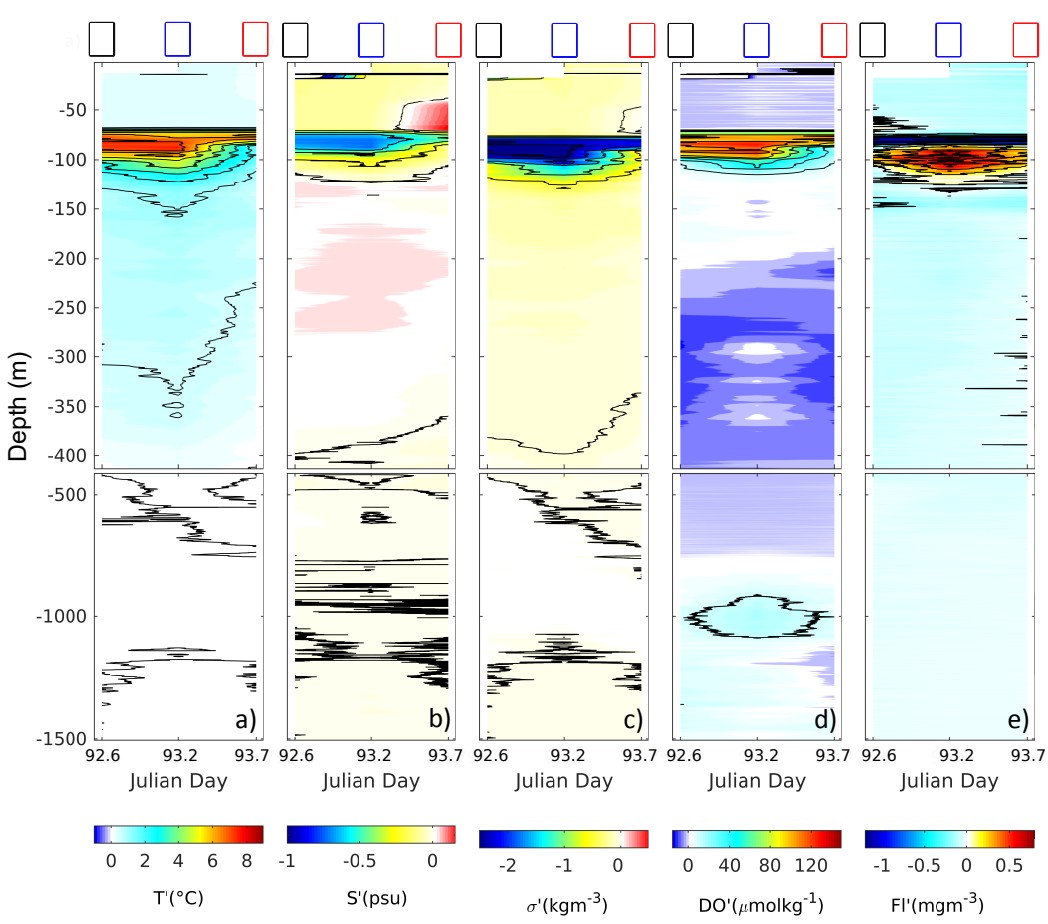

**Figure 6.** Vertical sections of a) temperature, b) salinity, c) density, d) dissolved oxygen, and e) fluorescence anomalies along the transect (A2-A3) from the center of the eddy to the western rim (CTD 08, 09, 10) with respect to a no-eddy situation shown as CTD 11 observation. The color boxes at the top of each subplot indicate the respective CTD stations as indicated in Table 1.





**Figure 7.** Current velocity at 40 m and 200 m water depth as the ship was crossing the northeastern eddy segment to the eddy center (transect A1-A2), and then from the eddy center to western eddy rim (transect A2-A3). The largest vector corresponds to a velocity of 45 $\mathrm{cm\,s^{-1}}$. White circles show the positions and times of stations at which corresponding CTD profiles were carried out. The contour plot in the background shows SLA on 02 April 2019.





**Figure 8.** a) Vertical structure of seawater temperature and b) salinity at a box with size of 1° by 1° at 80 m, 700 m, 1500 m, and 3000 m depths based on reanalysis data on 03 April 2019. Black arrows show the geostrophic flow at each depth with the largest arrow correspond to current speed of 50 $\mathrm{cms}^{-1}$. Color scale is different per depth interval to allow better distinction of features. Black ellipsoid indicates the eddy perimeter

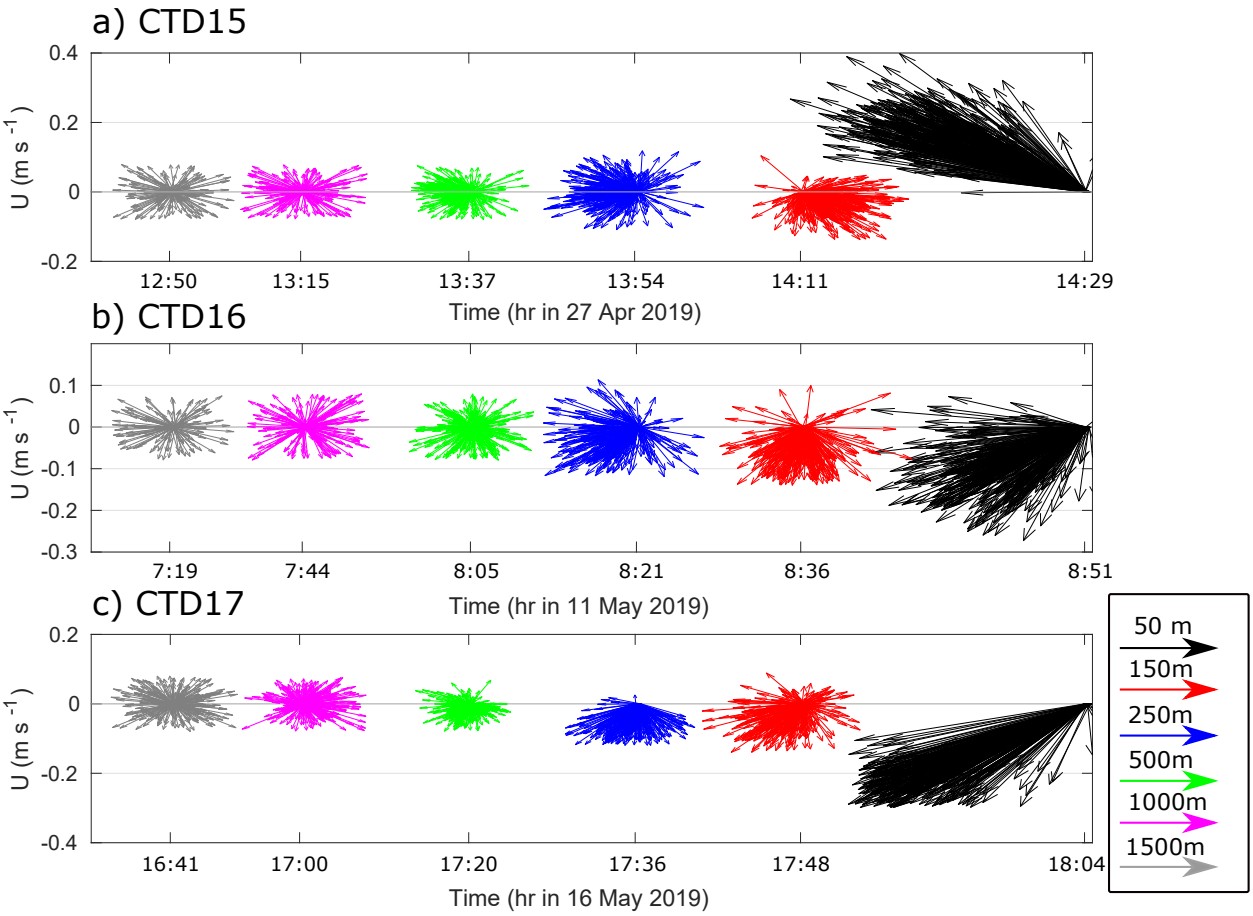

**Figure 9.** Current velocity obtained at the western rim, eddy center and eastern rim of the taken at different depths from an Aquadopp attached to the CTD at a) 27 April 2019, b) 11 May 2019 and c) 16 May 2019. Different colors correspond to the measurements at distinct depths.

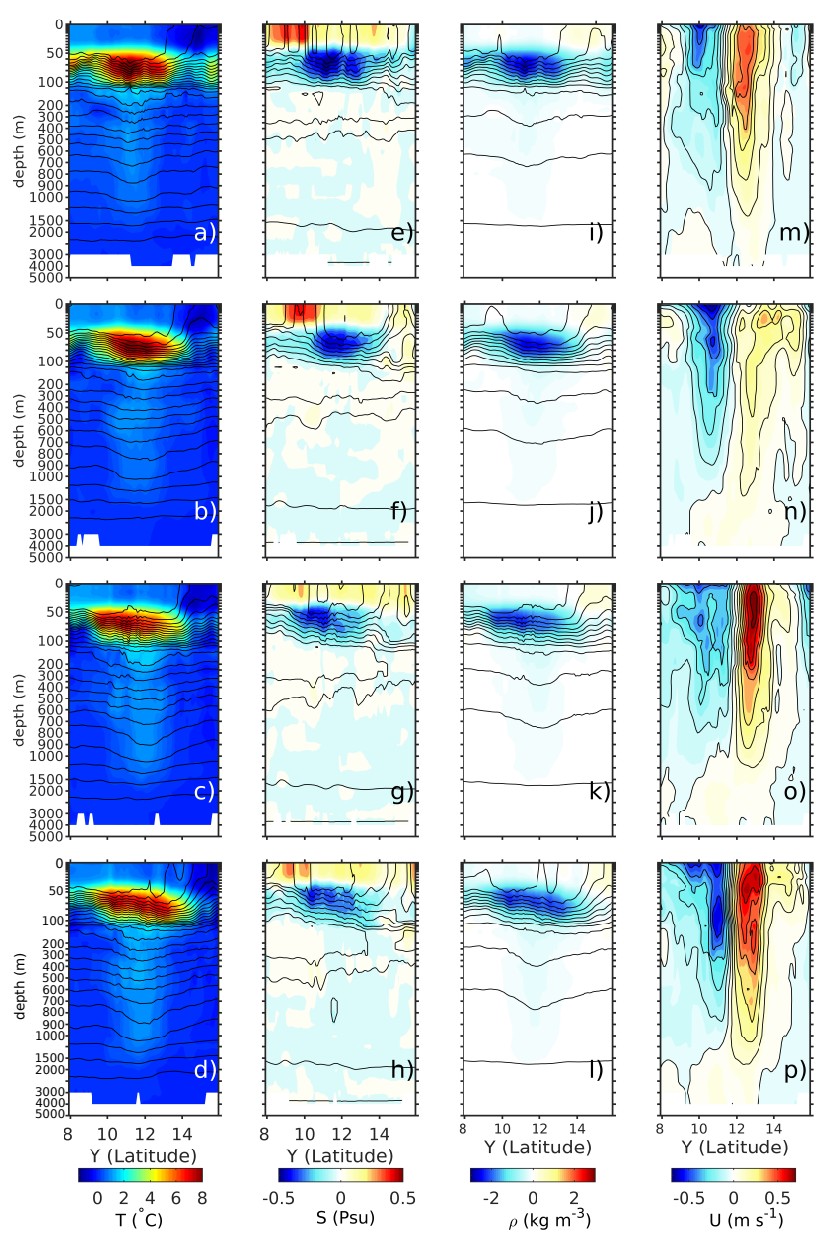

**Figure 10.** Vertical sections of a) to d) temperature, e) to h) salinity, i) to l) density and m) to p) zonal velocity at 03 April, 27 April, 11 May and 16 May 2019 (solid black lines) and their anomalies with respect to climatological mean values of 2018 (color shading). The solid lines in temperature field range between 2°C and 27°C in 1°C intervals. The solid lines in salinity field range between 34.7 and 33 with 0.2 psu intervals. The solid lines in density fields range between 1021 and 1027 $\mathrm{kg\,m^{-3}}$ in 0.5 $\mathrm{kg\,m^{-3}}$ intervals and the current field anomalies range between 0.7 and 0.7 $\mathrm{m\,s^{-1}}$ in 0.05 $\mathrm{m\,s^{-1}}$ velocity intervals.

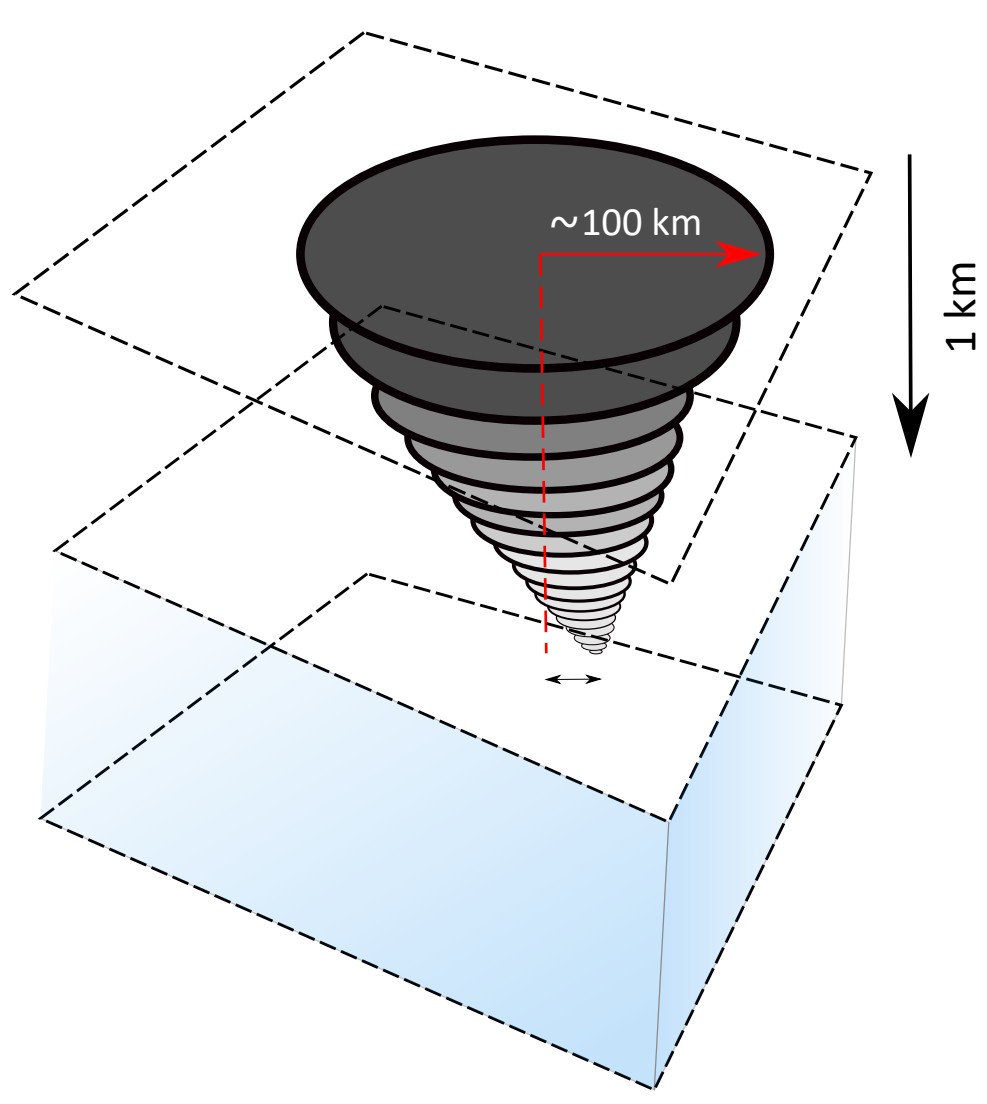

**Figure 11.** Schematic illustration of eddy in the water column based on reanalysis products



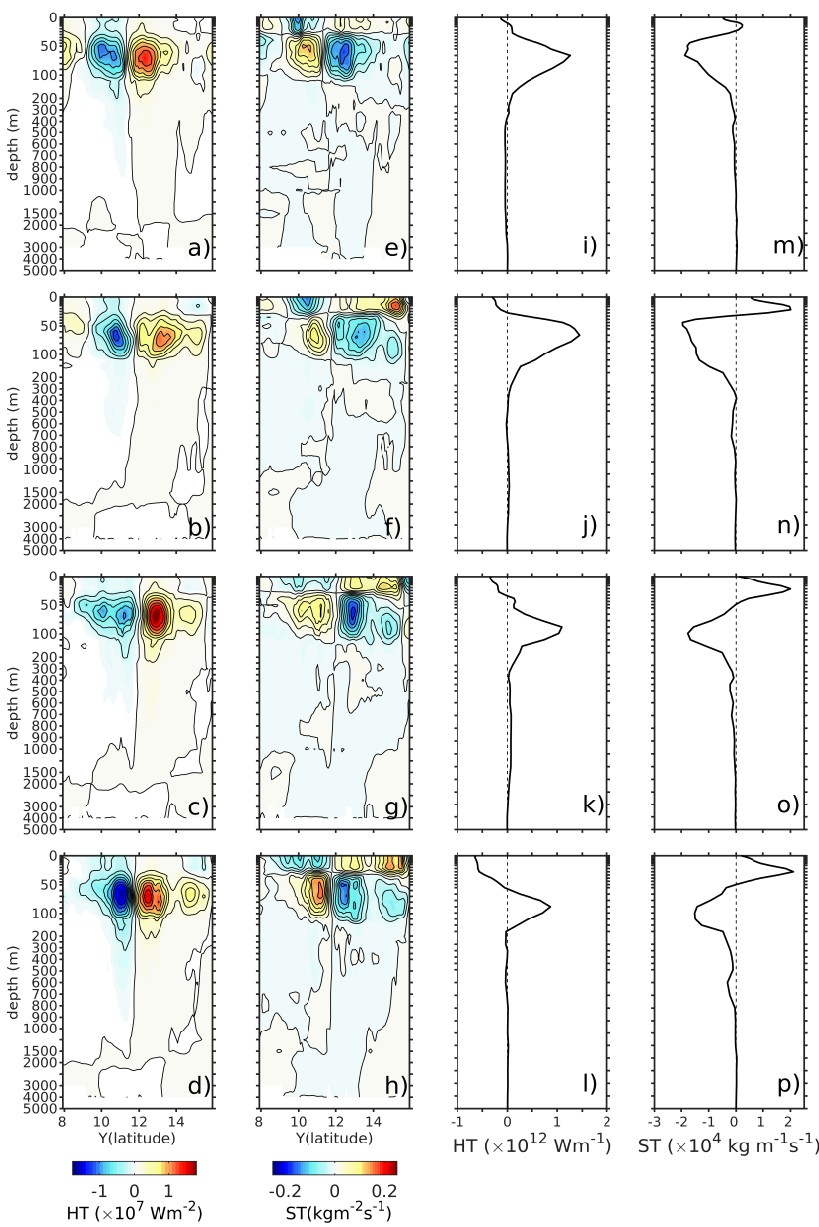

**Figure 12.** Vertical sections of heat flux a) to d) and salinity flux e) to h). Vertical profiles of heat transport integrated across the eddy area i) to l) and vertical profile of heat transport m) to p). Each row from top to bottom show the transects in 03 April, 27 April, 11 May and 16 May 2019 respectively.