# Peer review of "Impact of a long-lived anticyclonic mesoscale eddy on seawater anomalies in the northeastern tropical Pacific Ocean: A composite analysis from hydrographic measurements, sea level altimetry data and reanalysis model products"

_Ocean Science, 2022_

## Referee Comment (RC2)

[referee-annotated manuscript omitted]

---

## Author Comment (AC1)

**Responses to reviewer #1**

We thank the reviewer for carefully reading our manuscript and helping us to clarify the presentation of our results. All corrections, modifications and explanations are given in red color lines in the text of our manuscript as well.

In the Abstract authors presented the major outcome of this research – quantified anomalies, associated with one of the eddies, which authors remotely traced since June 2018 from its origin site in Tehuantepec Gulf over 2000 km away to the location of their in situ survey in April-May 2019. Alongside the anomalies values authors should include here the time-span period over which the mean was evaluated and the anomalies standard deviations. In the last sentence about the implications, the authors should clarify why an eddy 'traceable down to 1500 m' (Page 1, line 7) is important for the expected mining activity at a depths of 4-5 km in the area. Perhaps bringing in the ideas from lines 97-105 (Page 3) or from Conclusions could be helpful.

It is corrected in the Abstract as below.
Compared with annual climatological averages in 2018, the water trapped within the eddy are …

The last sentence is slightly corrected as:
Understanding the dynamics of strong mesoscale eddies that can affect the seafloor in this region of the Pacific Ocean is especially important in the light of potential deep-sea mining activities that are being targeted on this area.

In the Methods section authors briefly described the approach, however &2.4 requires stronger justification of the 'climatology' chosen to estimate the anomalies, since the suggested source is different from more traditional 1/4° fixed depths decadal WOA (Boyer, Levitus et al, 2019), isopycnal WAGHC (Gouretski, 2019) or several other monthly ocean climate products.

We have added the WOA18 climatology data for a more robust comparison in our study at section 2.4. In the revised version all observations are compared against the WOA18. Please see fig 5-6-7 and the corresponding text.

It is not very clear whether the authors used (a) an annual mean over whole 27-years reanalysis period (1993-2019) or (b) the multi-year monthly mean values at each grid cell (p.6, L. 150-151) derived from certain CMEMS GLORYS12 dataset. Authors applied individual daily reanalysis snapshots of the T,S etc properties interpolated/ or derived/ from the nearest grid cells to the location of their CTD stations in R/V Sonne 268 cruise. How were the reanalysis T,S profiles 'validated against them' (versa CTDs)? What bias ranges at what levels were detected (and exposed as shadows at Figure 4)?

The figure 5 is replotted using WOA18 (covering the period between 2005-2018). The shaded area is derived from standard deviation obtained over the long time-series (13 years). The reanalysis model validation against our in-situ observations is presented in an additional figure in the ms.

It seems appropriate to use the reference to relevant peer-reviewed papers (in addition to Tech. reports, i.e. Lellouche et al. 2021, https://doi.org/10.3389/feart.2021.698876). The latter could assure audience in the reproducibility of the results and increase the confidence in the quality of the products being used, which has its known uncertainty and well defined biases.

The reference is added to the study.

Results section provide convincing evidence of substantial differences induced by eddy transition into ambient ocean properties. Indication of which criterion (potential density or thermal) and which thresholds value were chosen to determine the mixed layer depth in &3.3, L.233 is required. Referenced Montegut et al 2004 shows wide spread in results with different threshold values applied ($\Delta T$=0.1, 0.2°C, $\Delta\sigma\theta$ = 0.03-0.055 kg m3 etc). What was the contribution in the uncertainty induced by barotropic motion (tidal, inertial) on vertical fluctuations of the thermocline and other isolines depths derived from CTD casts at the eddy centre and its opposite edges (L.234-236)?

We aware of such uncertainties in our study. However, with available measurements specially with very short temporal resolution in which a full tidal period did not cover, analysis the effect of barotropic motions on vertical fluctuation of thermocline is not possible.

Measured enhanced swirl/transition velocities ratio (Chelton, 2011) exceed 3 (L.274), therefore this particular eddy was capable in the net volume transport. Original author's method (subtraction of climatology / reanalysis profiles from the measured CTD values) enables to produce and to analyze the required estimates with adequate confidence and with known limitations. These results are well presented with good illustrations in paragraphs 3.5 and 3.6. The only exception is the clarity in description of the Aquadrop (current meter, mounted on CTD frame) horizontal velocity measurements (P.11, L.321-328) and Figure 9. More informative and consistent picture could be revealed with either (a) similar (quiver plot) vectors averaged over 2-5 minutes intervals or/& (b) its replacement with i.e. 'wind rose' diagram. The later will provide a percentage of the currents flowing toward each directional segment and the bar lengths proportional to velocity magnitude - at each depths level. It would be beneficial to re-arrange all 6 plots vertically for three (west, central and eastern) stations for better visualization.

As most of the CTD stops and the corresponding measurements cover only a time period between 5 and 10 mins, the suggested average period (2-5 mins) is too long for our measurements. So, we stick to the previous presentation of our data with the suggested method of wind-rose diagram. Other suggestions are taken in style of plotting figure. The figure 9 is replotted as suggested for wester, center and eastern location of the eddy.

In the Discussion section authors put the main findings of this research into essentially wider context. Promising advantage of the chosen method is an enabling to filter out the impact of a long-term oscillations such as ENSO. However, in parallel, provision of local anomaly values may be useful for comparison with eddies known from the literature. Abnormally large thermal anomaly (+7.8°C) here was similar to only one reported occasion elsewhere - in the South Chinese Sea, Chu et al 2014, who also used climatology instead of nearby background station(s). Authors reasonably explained their exaggerated T anomaly with several factors, including the longer than usual residence time of given eddy at the origin site, confirmed by the satellite data analysis. Coincidently that overlapped with positive (warm) ENSO phase in 2018-19 and with the downward descending of the warmer surface water due to anticyclone clockwise rotation within the eddy.

Thank you for your comment. We believe that it should be highlighted that the exaggerated T nomaly in our study is also confirmed by taking long-term reanalysis model as the reference for the comparison and similar large anomalies were depicted again.

The high practical value of the study is in confirming the possibility of remote detection well in advance the arrival of mesoscale eddies at the target areas. Study proves the importance of eddies dynamics in enhancing the transport of heat, salt, dissolved and suspended matter in a belt overlaying the perspective mining sites in the NETP.

**Technical corrections**

Page 1, line 11. Is there word missing: salt transport [anomalies] or the given value is 'absolute' transport with the eddy?

Corrected in the text.

Page 1, line 19. Is there word missing:' long distance [horizontal]'.

Added to the text.

Page2, line 27. Better replace the word 'hydrography' with "dynamics" or similar, as term 'hydrography' is the definition of the mapping, surveying and charting the water bodies, rather than changing the sea-water properties or ocean motion.

Corrected in the text.

Page 4, lines 99-100. Replacement one of two 'potentially' occasions with 'perhaps' or similar.

Replaced.

Page 8, lines 190-210, &3.2 and Figure 4 (Page 23). Adding the Water Mass indices (with small symbols or bounding lines) on top of the T, S profiles and/or T,S diagram (Figure 4a,b) will provide the reader with better context about where-in the given eddy's profiles are embedded. Changing the axes of T,S diagram with equal size could help to avoid WM indexes overlaps. Adding dashed grid to Fig.4a and depths marks (0.2, 0.4,1 – 4 km) at Fig.4b could be helpful.

The figure is replotted based on WOA18 and the water mass indications are added to the subplot a).

Page 12, line 365. Replace 'these' with 'the' at: 'in these other regions'.

Done.

Page 14, line 420. The eddy-induced variations in the depth-integrated heat transport profile order of O (+10-100 TW) here contrast with the values +26-150TW included elsewhere in the text (Page 12, L.363) and in the Abstract +85 TW (L.10). It seems reasonable to complete integration calculus and provide a volume integral value as well, i.e. using a formula for tilted conus volume (Fig. 11, or other approximations) and with the STD spread due to eddy shape uncertainty. Similar is applicable to Salt volume transport.

The text is corrected. Based on the suggestion, we present it based on a mean and a standard deviation. See line 475.

Page 14, line 421. How did authors obtain the value of 'the annual meridional heat at 10°N'? That needs description or reference to the source.

This has been addressed with editing corresponding text and new references.

Page 22, Figure 3, include position of CTD stations at Fig. 3b,c,d

The figure is modified based on the suggestions. The location of all CTD casts are added to the figure 4.

Page 23-24 Figures 4a and 5 could become much more informative if the different vertical and horizontal axes limits and un-even stretching would be applied to the upper and to the deeper layers, i.e. below 0.5 km. That is possible in the same manner as authors did on Figures 6, 10, 12 (pages 25, 29, 31).

Figure 5 and 6 are modified based on an uneven vertical axis.

Page 27, Figure 8. Increase the font size of the tick marks (colorbars, axes) will make it visible (size 14-16 points). It is sufficient to show once the value of tick-marks on geographical axes here. Colour scheme could be unified for all layers and shown once – if it's log-stretched at both ends (i.e. as shown at Fig.4 in Chu et al 2014, referenced here).

Figure 8 is replotted based on your suggestion and the ticks are adjusted. Please see Fig 12.

Page 28, Figure 9 – see the notes in Specific comments on wind-rose diagram and/or vectors averaging over 2-5 minutes intervals.

Figure 9 is adjusted based on the suggestions and replotted at Fig 10.

---

## Author Comment (AC2)

**Response to reviewer #2**

We thank the reviewer for carefully reading our manuscript and helping us to clarify the presentation of our results. All corrections, modifications and explanations are given in red color lines in the text of our manuscript as well.

Hu et al., 2018?

It is corrected to He at al.

Analysis of altimeter data shows that the secondary maximum wind intensity in the TT Gulf formed a long-lived ACE in this region between 15 and 23 July 2018. please provide a more direct evidence on this conclusion.

The figure shows wind speed in the TT gap wind region from January 2018 to August 2019 (data are from the ECMW database). The secondary maximum wind velocity in summer is not a very frequent event, but in July 2018 there was a relatively strong wind event over a period of about 16 days. This eventually led to eddy formation during the summer season.

[Figure]

The wind velocity variation at Tehuantepec Gulf is shown in black. A moving average with 7-day interval is shown in red.

It's better to add a supplement movie demonstrating the formation and evolution of the eddy.

A video showing the eddy genesis and its evolution is added to the appendix.

Considering the large uncertainty in model simulations, I suggest the authors to use the WOA18 products or climatological averages of WOD profiles as referenced large-scale background values.

We added the WOA18 to our analysis, specially for the comparison of T-S diagram with in-situ observations and analysis of all CTD observations. Please see the section 3.3 and 3.4 for the changes using WOA18 as the reference. However, due to the absence of current velocity data in WOA18, we could not further use WOA18 in our analysis. Therefore, for the sake of consistency we stick to reanalysis model products for the HT/ST. A validation for reanalysis data against in-situ observation is shown and added to the ms.

Temperature, salinity, and oxygen anomalies induced by the eddy The vertical temperature (T), salinity (S), and dissolved oxygen (DO) profiles obtained from an E-W transect across the eddy center (Figure 1b) measured within 2 days from 02 April to 04 April 2019 are shown ….
It is difficult for me to distinguish which CTD profile was in the eddy. The authors may want to mark the relative positions of the CTD profiles to the eddy first in Figure 2 or Figure 3.

The relative position of measurements to the eddy is illustrated in Figure 4. The text is rewritten in lines 115-125 for a better clarification. All CTD stations and their relative position to the eddy is also given in the Table 1. The eddy appearance at each station is explicitly given in an extra column for better clarifications.

The authors may, firstly, add a comparison of the evolutions of SLA and SST between satellite measurements and the reanalysis data, as well as a movie of the evolution of 3D Temperature structure at the eddy positions in the reanalysis data, to confirm that the eddy was successfully reproduced by the model (Figures 4-6 in He et al., 2017 may be a good reference).

To show that the reanalysis product successfully produced the 3D structure of eddy, we illustrate the comparison of reanalysis-model products with the in-situ observations of T and S.  The comparison of in-situ observation and reanalysis model in capturing the T/S anomalies also clearly shows that the model successfully reproduced the eddy impact (Figure 7 and 11).
In addition, the northeastern tropical Pacific is located in the warm pool of the Western Hemisphere, which is very homogeneous in the SST. The warm pool grows especially in April and May in this region, with this regime reaching its maximum development in the open ocean. This makes the detection of a warm eddy-driven T anomaly in this region, which mostly occurs at subsurface region, a very difficult task (Wang, C. and D.B. Enfield, 2001: The tropical Western Hemisphere warm pool. Geophys. Res. Lett. 28: 1635-1638). The 3D temperature structure of the eddy is being revised in the ms with the SSHA depicted at the sea surface (shown as white contours). Please see Fig. 12.

Line 290, 700?

Corrected to 700 m.

Line 292, north-south section

Corrected

Same line, anomalies across the eddy center

Corrected

Line 293, values

Corrected

This Figure should be placed before the Figure 9.

The figure presence re-adjusted.

How was the "trapping depth" estimated here?
The trapping depth is set to 1500 m where the ultimate T anomaly were observed.

These are model results rather than observations.
The text is corrected.

Interestingly, the positive S anomaly is observed in all profiles from the ocean surface to 35 m depth (Figure 10e-h). Again, these are not observational results. compared with the CTD profiles in Figures 5 and 6, these positive S anomalies are more likely model errors instead.

The anomalies calculated with the comparison of our observation with WOA18 confirm that the positive S anomaly illustrated in the reanalysis model are indeed correct (Fig 7b). Using WOA18 data was a great point to clear this uncertainty in the previous analysis.

Fig 9 in line 321, This Figure should be placed before the Figure 10.
The figures presence is adjusted.

Line 321, northwestward?

It is removed from the text.

Line 351, meridional?
It is corrected to zonal.

Line 358, Given the authors presented the temporal evolution of the HT in Figure 12, readers may be more interested in why the HT reached the peak at 27 April.

The maximum HT is derived from maximum T anomaly observed at 27 April. This could be due to the different background seawater temperature available at the various transects. The transects are taken at a spatial distance of about 5° from 113°W and 118°W which indicates that the different water mass can occur.

Line 363-365 Please make sure that you are comparing zonal heat\salt transports by the rotation of the eddy with meridional eddy transports in those studies.

We present a new comparison with previous studies in the text. And the heat/salt transport are entirely edited in the text to meridionally-integrated zonal transport. See text lines between 400-420

Line 417, the thermocline?
It is added to the text.

Line 421, It is not appropriate to compare zonal transports with meridional transports.
The comparison with meridional T/S transport in previous studies are removed from the ms. Some additional references are replaced. Please see lines 405-425.

this study has shown that this may not be the case for all mesoscale ACEs passing through the NETP. Only a eddy case cannot reach this conclusion.

The text is accordingly edited. Whereas some strong mesoscale eddies passing over the area targeted for deep-sea mining change the near-bottom current regime at depths >4000 m and potentially may affect the deep seafloor and its ecosystems (Aleynik et al., 2017; Purkiani et al., 2020), weaker

mesoscale eddies may only affect the properties of seawater in the upper ocean layers without affecting the hydrodynamics of the seafloor.

Salt?
In figure 13, the text (Salt) in corrected.

---

## Author Response (AR2)

**Response to Reviewer #2**

The suggested corrections and improvements are included in the final version as below:

You have demonstrated the existence of the wind event in July 2018 at the TT Gulf, but the supplementary movie seems to start from August 2018. Is it possible to start the movie before July 2018 such that the effect of the wind event can be seen?

A new video of the eddy genesis in the region starting from 01 June 2018 is produced.

L.5 -114 → 114 -- remove the minus sign

It is removed and corrected in the text.

L.31 EDA → eddy detection algorithm (EDA) -- first appearance in the main text

It is added in the main text.

L.82 "could resulted" -- please fix.

The typo is corrected.

L.87 "(Nishino et al., 2011)" shoud be without brackets.

The citation is corrected.

L.177 What are the "ultimate" temperature and salinity?

The sentence is rephrased as:

In contrast to assuming a depth of no motion at 1200 m (e.g. He et al. (2018)), heat and salt transport were calculated by integrating equations 4 and 5 for the upper 1500 m water depth, where the sign of the temperature anomaly changes.

L.185 "consistently mimicked" -- do you mean "reproduced" (in the model)?

"reproduced" is replaced in the text.

Fig.11 caption Salinity between 34.7 and 33 cannot be contoured at the interval of 0.2

**because 34.7-33=1.7 is not a multiple of 0.2.**

It is corrected to: between 33 and 34.8